# THE ROLE OF DEDUCTIVE AND INDUCTIVE REASONING IN LARGE LANGUAGE MODELS

## ABSTRACT

Large Language Models (LLMs) have achieved substantial progress in artificial intelligence, particularly in reasoning tasks. However, their reliance on static prompt structures, coupled with limited dynamic reasoning capabilities, often constrains their adaptability to complex and evolving problem spaces. In this paper, we propose the **D**eductive and **InD**uctive(**DID**) method, which enhances LLM reasoning by dynamically integrating both deductive and inductive reasoning within the prompt construction process. Drawing inspiration from cognitive science, the DID approach mirrors human adaptive reasoning mechanisms, offering a flexible framework that allows the model to adjust its reasoning pathways based on task context and performance. We empirically validate the efficacy of DID on established datasets such as AIW and MR-GSM8K, as well as on our custom dataset, Holiday Puzzle, which presents tasks about different holiday date calculating challenges. By leveraging DID's hybrid prompt strategy, we demonstrate significant improvements in both solution accuracy and reasoning quality, achieved without imposing substantial computational overhead. Our findings suggest that DID provides a more robust and cognitively aligned framework for reasoning in LLMs, contributing to the development of advanced LLM-driven problem-solving strategies informed by cognitive science models.

## 1 INTRODUCTION

*"The measure of intelligence is the ability to change."* – Albert Einstein

Large Language Models (LLMs), such as GPT-4, have transformed natural language processing by excelling in tasks such as language translation, summarization, and question-answering (OpenAI, 2023), particularly in reasoning tasks and few-shot learning. However, there is ongoing debate regarding their problem-solving reliability. According to Zhou et al. (2024), scaling up and fine-tuning LLMs enhances their capabilities but also diminishes reliability, introducing unpredictable errors even in simple tasks and reducing the effectiveness of human supervision. Conversely, Li et al. (2024) highlights that the application of the Chain of Thought (CoT) (Wei et al., 2022b) methodology significantly improves the accuracy of LLMs in arithmetic and symbolic reasoning tasks by enabling inherently serial computations, which pose challenges for low-depth transformers. Furthermore, Bubeck et al. (2023) observes that LLMs demonstrate a high degree of accuracy and consistency in multi-step reasoning tasks, particularly when employing techniques such as CoT and self-consistency (Wang et al., 2022). Additionally, reinforcement learning from human feedback (RLHF) has been shown to enhance model performance, notably reducing the incidence of harmful or inaccurate outputs (Ouyang et al., 2022; Christiano et al., 2017). These insights suggest that, despite concerns related to model scalability and the potential for errors introduced during fine-tuning (Zhou et al., 2024), LLMs can exhibit considerable reliability in complex reasoning tasks when guided by structured methodologies and reinforced with human feedback. Ensuring the robustness of LLM outputs remains a critical priority, necessitating further investigation into strategies aimed at enhancing model resilience and dependability.

Despite the notable success of LLMs, they face several limitations when dealing with more complex and evolving tasks. In particular, their rigidity in reasoning and difficulty in generalizing across diverse problem types present significant challenges. A key limitation of current LLMs is their

reliance on static prompt structures and patterns learned during training, which restricts their adaptability in novel or evolving contexts. These models often apply fixed strategies to problem-solving, leading to challenges in tasks that require logical reasoning, such as calculating family relationships, performing numerical comparisons, or counting specific characters in a word (Nezhurina et al., 2024). Although these tasks may seem straightforward, LLMs tend to depend on pre-learned patterns instead of dynamically adjusting their reasoning processes, resulting in errors in more complex problem spaces (Marcus, 2020; Hendrycks et al., 2020). This inflexibility contrasts with human problem-solving, which is typically iterative and adaptive (Sloman, 2009). Humans use inductive reasoning to derive general rules from specific instances and then apply deductive reasoning to novel situations, allowing for dynamic strategy adjustments based on task complexity. In contrast, current LLMs lack this level of flexibility in reasoning, which limits their ability to generalize and adapt to more sophisticated scenarios. This underscores the need for increased attention and research in this area.

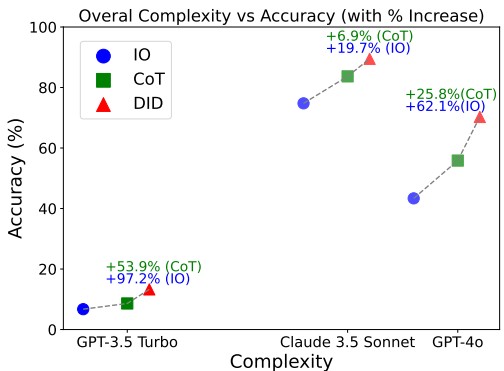

Figure 1: Comparison of reasoning approaches in LLMs including the IO, CoT, and DID framework, highlighting the progression from direct output generation to dynamic inductive and deductive reasoning for more adaptive problem-solving.

Moreover, LLMs have difficulty generalizing reasoning across tasks that require dynamic adjustment or incremental problem-solving. While LLMs can achieve high accuracy on specific tasks, their performance often degrades when confronted with problems that evolve or require multi-step reasoning. This issue is especially evident in tasks where the model must balance different reasoning strategies or integrate information from multiple sources. Tasks such as stepwise numerical reasoning, temporal reasoning, or complex multi-step inference highlight gaps in LLMs' ability to maintain consistent reasoning across different stages of a task. These models tend to produce inconsistent or contextually inappropriate answers when required to adjust their reasoning dynamically as the problem unfolds. The static and inflexible nature of their reasoning pipeline limits generalization and accuracy, particularly when compared to human problem-solving, which adapts to new information in real-time. Despite improvements with techniques like CoT (Wei et al., 2022b), Tree-of-Thought (ToT) (Yao et al., 2024), Temperature-Tree-of-Thought ($T^2oT$) (Cai et al., 2024), and Graph-of-Thought (GoT) prompting (Besta et al., 2024), current LLMs still struggle to adjust their reasoning dynamically, resulting in difficulties in addressing more fluid and complex tasks.

To address these challenges, we propose the De-In-Ductive (DID) method, a novel approach designed to enhance LLM reasoning by integrating both inductive and deductive reasoning processes within the prompt construction framework as the Figure 1 shows. Grounded in cognitive science models of human reasoning, the DID method enables LLMs to adjust their reasoning pathways dynamically in response to the task context and its evolving complexity. In the DID method, inductive reasoning is first used to derive general rules from specific instances, followed by deductive reasoning to apply these rules in solving particular problems. This hybrid reasoning process mirrors human cognitive strategies, allowing the model to adjust its reasoning dynamically based on real-time feedback. By employing this dynamic prompt strategy, the DID method improves the adaptability and flexibility of LLMs, enabling them to better handle complex, evolving problem spaces.

We validate the effectiveness of the DID method on established benchmarks such as AIW and MR-GSM8K (Wei et al., 2022a; Cobbe et al., 2021), as well as our custom dataset, Holiday Puzzle, which includes tasks about holiday date calculations. By leveraging DID's hybrid prompt strategy, we observe significant improvements in both solution accuracy and reasoning quality, achieved without imposing substantial computational overhead. These results demonstrate the efficacy of DID in addressing the limitations of current LLM approaches. This work provides the following key contributions:

- We introduce a De-In-Ductive (DID) methodology that integrates both inductive and deductive reasoning within LLM prompt construction. This dynamic approach addresses key limitations of static prompt structures, enhancing the model's reasoning flexibility and adaptability.

- Through empirical evaluations, we demonstrate that the DID method significantly enhances the adaptability and efficiency of LLMs across a diverse set of complex tasks. Moreover, DID improves solution accuracy and reasoning quality without incurring substantial computational overhead.

## 2 RELATED WORKS

**Cognitive Science and Deductive-Inductive Reasoning** Deductive and inductive reasoning are foundational concepts in cognitive science for understanding human thought processes. Deductive reasoning, formalized by philosophers like Kant, involves applying general principles to specific cases, ensuring conclusions logically follow from premises. Inductive reasoning, conversely, generalizes from specific observations to form broader conclusions, as highlighted in *The Riddle of Induction* (Goodman, 1983). Cognitive models view these modes of reasoning as complementary, where inductive reasoning generates hypotheses, and deductive reasoning tests them (Wason, 1960). The interplay between these two reasoning methods has been shown to enhance problem-solving accuracy, particularly in uncertain domains where balancing exploration and validation is critical (Johnson-Laird, 1983; Kahneman & Tversky, 1974). Research on mental models and heuristics highlights how this dynamic combination allows for more flexible reasoning, especially in tasks characterized by complexity or ambiguity. Problems in cognitive science are often classified based on their structure and uncertainty: well-structured problems (e.g., mathematical proofs) lend themselves to deductive reasoning, whereas ill-structured or open-ended problems (e.g., scientific discovery) require inductive reasoning to form plausible hypotheses from incomplete data (Funke, 2013). Cognitive insights have been increasingly integrated into neural networks (L Griffiths et al., 2008), with recent studies emphasizing the importance of embedding inductive structures within models to improve generalization across tasks (Tenenbaum et al., 2011). The DID framework builds on this cognitive science foundation by dynamically combining inductive and deductive reasoning within LLMs, creating a hybrid model that mirrors human cognitive processes and enhances adaptability in problem-solving.

**LLMs for Reasoning and Prompting Techniques** While LLMs like GPT-4 have shown remarkable capabilities in tasks such as text generation and summarization (Brown et al., 2020), they often struggle with structured reasoning, particularly in tasks involving logical inference, numerical comparison, and complex deduction (OpenAI, 2023; Nezhurina et al., 2024). These shortcomings have been well-documented in challenges like the ARC Prize (Rae et al., 2021) and in tasks that require step-by-step reasoning or multi-hop inferences. To address these limitations, prompting techniques such as CoT prompting (Wei et al., 2022b),ToT (Yao et al., 2024), and GoT (Besta et al., 2024) have been developed to improve LLMs' capacity for structured reasoning. CoT enhances stepwise reasoning by breaking down complex problems, while ToT and GoT explore multiple solution paths through structured thought representations. However, these approaches remain static, requiring extensive prompt engineering and lacking the dynamic adaptability needed for diverse tasks. Recent work, such as Hypergraph of Thoughts (HoT) (Yao et al., 2023), extends these methods to multimodal and more complex reasoning but still fails to offer the real-time adaptability needed for nuanced problem-solving. Another critical analysis by Marcus (2020) highlights the limitations of current LLMs, emphasizing their struggles with maintaining logical consistency and coherence in complex reasoning tasks. The DID framework addresses these gaps by dynamically integrating inductive and deductive reasoning, making the reasoning process more adaptive and context-sensitive, informed by the probabilistic reasoning advancements seen in the combination of Bayesian models with neural networks (Gershman et al., 2015).

## 3 METHODOLOGY

The De-In-Ductive (DID) framework dynamically integrates inductive and deductive reasoning, inspired by cognitive science models of human reasoning. It balances inductive hypothesis generation

and deductive rule application to improve the flexibility and adaptability of large language models (LLMs) in complex problem-solving tasks.

## 3.1 PRELIMINARY

In this section, we formalize the core assumptions underlying the DID framework, which define the problem space and establish the performance limitations of LLMs in solving these problems. These assumptions lay the theoretical groundwork for modeling the interaction between problem complexity and the reasoning capabilities of LLMs.

**Assumption 1 (Problem Distribution and Complexity)**    Let $\mathcal{P}$ denote a problem space, where each problem instance $p \in \mathcal{P}$ is associated with an observed dataset $D_p = \{d_1, d_2, \ldots, d_n\}$ and additional new data $D_{\text{new},p} = \{d'_1, d'_2, \ldots, d'_m\}$. Each problem $p$ is governed by a true underlying hypothesis $H_p$, which explains the relationship between the data and the problem context.

The observed data $D_p$ follows a joint distribution conditioned on the hypothesis $H_p$:

$$P(D_p \mid H_p) = \prod_{i=1}^{n} P(d_i \mid H_p), \tag{1}$$

where $d_i$ is independently generated according to the true hypothesis $H_p$. During the deductive phase, the model tests a candidate hypothesis $H$ against the new data $D_{\text{new},p}$, and the likelihood is given by:

$$P(D_{\text{new},p} \mid H) = \prod_{j=1}^{m} P(d'_j \mid H). \tag{2}$$

The complexity of each problem $p$ is characterized by a parameter $c(p) \in \mathbb{R}_+$, where higher values of $c(p)$ indicate more complex problems, influencing the difficulty of hypothesis testing and data modeling.

**Assumption 2 (Baseline Performance and Deductive Probability)**    Let $\mathcal{M}_{\text{LLM}}$ represent a baseline LLM with parameters $\theta$. For problems with complexity $c(p) \geq c_0$, the likelihood that $\mathcal{M}_{\text{LLM}}$ produces a correct solution is bounded by:

$$\mathbb{P}_{\text{correct}}(p, \theta) \leq \epsilon, \quad \text{for } c(p) \geq c_0, \tag{3}$$

where $\epsilon \ll 1$ reflects the baseline model's limitations on novel or complex tasks.

During the deductive reasoning phase, the likelihood that the hypothesis $H$ holds, given the new data $D_{\text{new}}$, is computed by:

$$P_{\text{deductive}}(H \mid D_{\text{new}}) = \prod_{i=1}^{m} P(d'_i \mid H, \theta), \tag{4}$$

where the model tests the validity of $H$ using the new observations $D_{\text{new}}$ and adjusts its confidence in the hypothesis.

## 3.2 DE-IN-DUCTIVE (DID) FRAMEWORK

Figure 2 illustrates the comparison between the IO, CoT, and DID frameworks. The IO (Input-Output) Method processes natural language queries by retrieving patterns and facts without engaging in iterative reasoning. The Chain of Thought (CoT) Method improves logical reasoning by breaking down complex problems into sequential steps. Our proposed De-In-Ductive (DID) Method goes further by dynamically integrating inductive and deductive reasoning. By iteratively generating and testing hypotheses, DID adapts to problem complexities more effectively than static methods like

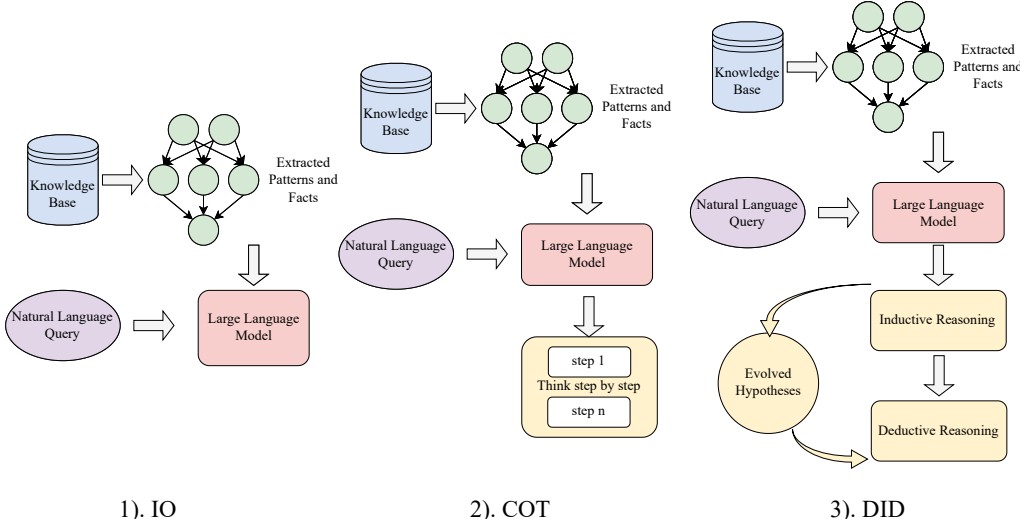

1). IO         2). COT         3). DID

Figure 2: Comparison of reasoning approaches in Large Language Models (LLMs) including the IO method, Chain of Thought (CoT) prompting, and the De-In-Ductive (DID) framework, highlighting the progression from direct output generation to dynamic inductive and deductive reasoning for more adaptive problem-solving.

CoT, optimizing problem-solving by balancing reasoning modes in response to task difficulty. Based on Assumptions 3.1 and 3.1, DID optimizes an objective function that balances both reasoning modes, dynamically adjusting as the complexity of the problem increases.

**Inductive Reasoning** The inductive phase generates hypotheses from observed data $D_p$. Using Bayesian inference, the posterior probability for a hypothesis $H$ is calculated as:

$$P(H \mid D_p) = \frac{P(D_p \mid H)P(H)}{P(D_p)},$$
(5)

where $P(H)$ is the prior probability and $P(D_p \mid H)$ is the likelihood of the observed data under hypothesis $H$.

**Deductive Reasoning** In the deductive phase, the generated hypothesis $H$ is tested against the new data $D_{\text{new},p}$. The likelihood of the new data given the hypothesis is:

$$P(D_{\text{new}} \mid H) = \prod_{j=1}^{m} P(d'_j \mid H).$$
(6)

This phase refines the hypothesis by comparing how well it explains the new observations.

**Hybrid Objective Function** The DID framework minimizes a hybrid objective function that integrates both inductive and deductive reasoning losses. The total objective function is defined as:

$$\mathcal{L}_{\text{DID}}(\theta) = \alpha \cdot \mathcal{L}_{\text{inductive}}(\theta) + (1 - \alpha) \cdot \mathcal{L}_{\text{deductive}}(\theta),$$
(7)

where $\mathcal{L}_{\text{inductive}}(\theta)$ represents the error in generalizing from the data, and $\mathcal{L}_{\text{deductive}}(\theta)$ penalizes errors in applying the rules to new instances. The weighting factor $\alpha$ adjusts dynamically based on the task complexity and the relative uncertainties in the inductive and deductive phases.

**Dynamic Adjustment**  The balance between inductive and deductive reasoning is dynamically weighted based on the uncertainties in each reasoning process:

$$\alpha_t = \frac{\mathbb{E}_H[P_{\text{inductive}}(H \mid D_p)]}{\mathbb{E}_H[P_{\text{inductive}}(H \mid D_p)] + \mathbb{E}_{D_{\text{new}}}[P_{\text{deductive}}(D_{\text{new}} \mid H)]}. \tag{8}$$

This adaptive mechanism ensures that DID can adjust its strategy according to the evolving complexity of the problem.

**Limitations of Chain-of-Thought (CoT) under Assumptions**  Under Assumptions 1 and 2, we infer that static reasoning methods such as Chain-of-Thought (CoT) prompting are insufficient for solving problems from the novel distribution $\mathcal{P}$. Since CoT relies on pre-learned patterns and fixed reasoning pathways, it lacks the dynamic adaptability required to handle new and complex problem structures. Specifically, without the ability to adjust $\alpha$ dynamically and integrate inductive hypothesis generation, CoT cannot effectively improve the low baseline performance ($\mathbb{P}(\text{Correct} \mid p, \theta) \leq \epsilon$) for complex problems.

**Computational Complexity**  The computational complexity of the DID algorithm can be expressed as $O(T \cdot (n_{\text{ind}} + n_{\text{ded}}))$, where $T$ is the number of iterations, $n_{\text{ind}}$ is the complexity of inductive reasoning per iteration, and $n_{\text{ded}}$ is the complexity of deductive reasoning per iteration. The dynamic adjustment enabled by the DID framework, as necessitated by Assumptions 1 and 2, reduces error propagation compared to static methods like CoT, thus improving efficiency and accuracy.

**Efficiency Gains**  By dynamically adjusting reasoning pathways according to task complexity and problem novelty, DID reduces the number of iterations required to converge on a solution. This adaptive process ensures that DID outperforms static methods like CoT in both performance and computational overhead, achieving higher accuracy with fewer computational resources.

**Theoretical adaption analysis**  The DID framework effectively manages cognitive load by initially focusing on simplified problem versions, allowing the model to concentrate on essential elements before engaging with more complex interactions. Through the inductive phase, the model observes specific instances, forming a foundation for generalization. As task complexity increases, the model transitions to deductive reasoning, applying generalized rules to arrive at a solution. This dynamic adjustment of reasoning strategies based on evolving task contexts enhances the model's adaptability and problem-solving efficiency.

**Integration with Existing Models**  The De-In-Ductive (DID) method is compatible with various LLM architectures and can be seamlessly integrated with existing techniques such as CoT prompting (Wei et al., 2022b). By providing a structured reasoning framework that dynamically incorporates both inductive and deductive reasoning, DID complements these methods and enhances the model's problem-solving capabilities. Specifically, it structures the problem-solving process to allow for the dynamic integration of reasoning strategies, utilizes a structured template prompt that guides the model through incremental reasoning stages and improves adaptability and problem-solving efficiency without introducing significant computational overhead. By mirroring human adaptive reasoning processes, the DID method provides a more flexible and robust framework for LLMs to tackle complex and evolving problems.

## 4 EXPERIMENTS

### 4.1 ALICE PROBLEMS

**Task.**  The AIW dataset is focused on evaluating the logical reasoning and deduction abilities of large language models (LLMs). The problems are structured around scenarios where models must infer relationships between family members based on a set of constraints, typically involving siblings, with questions like determining how many sisters or brothers a particular sibling has. Due to the AIW GitHub open-source dataset being provided in the form of questions and various prompts,

| Model\Prompt Method | Alice Problem | | | MR-GSM8K | | Holiday Puzzle | | |
|---|---|---|---|---|---|---|---|---|
| | IO (%) | CoT (%) | DID (%) | CoT (%) | DID (%) | IO (%) | CoT (%) | DID (%) |
| GPT-3.5 Turbo | 6.7 | 8.6 | 13.3 | 68.1 | 73.3 | 0.2 | 1.4 | 5.6 |
| GPT-4o | 43.4 | 55.9 | 70.3 | 82.0 | 83.7 | 7.8 | 5.2 | 15.4 |
| Claude 3.5 Sonnet | 74.8 | 83.7 | 89.5 | 91.3 | 92.0 | 17.4 | 17.8 | 24.5 |

Table 1: Merged Results for GPT-3.5 Turbo, GPT-4o, and Claude 3.5 Sonnet across Different Tasks (Alice Problem, MR-GSM8K, Holiday Puzzle)

we manually removed all prompts and eliminated duplicate questions that remained after prompt removal. This resulted in 113 unique original Alice problems. All subsequent experiments are based on these 113 problems, with results averaged over 20 runs.

**Baseline and Framework Setup.** We compare the performance of the DID framework with three other prompting methods: the IO prompt (which directly utilizes the LLM without structured prompting) and the CoT prompt. All methods are evaluated in a zero-shot setting. The comparisons are performed using three representative models: GPT-4o, GPT-3.5-turbo, and Claude 3.5 Sonnet. GPT-4o was selected due to its highest reported accuracy in the AIW paper, while Claude 3.5 Sonnet achieved the second-highest accuracy. GPT-3.5-turbo, by contrast, demonstrated mid-to-low accuracy. For a fair comparison, all model parameters, including temperature, top-k sampling, and other hyperparameters, are maintained at their default values.

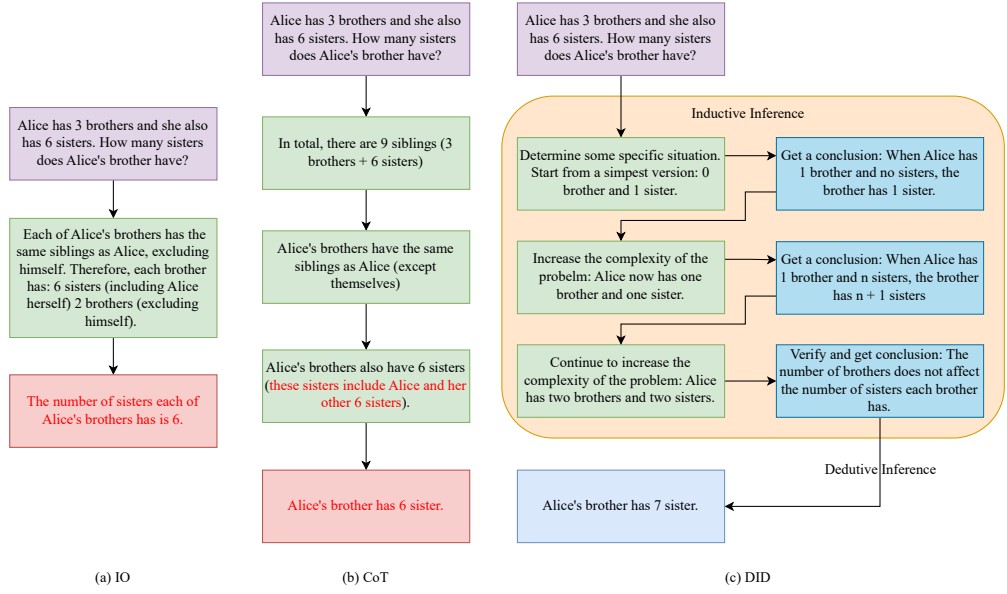

Figure 3: Comparison of reasoning approaches in Large Language Models (LLMs) including the IO method, Chain of Thought (CoT) prompting, and the De-In-Ductive (DID) framework, highlighting the progression from direct output generation to dynamic inductive and deductive reasoning for more adaptive problem-solving.

**Results** The results in 4 present a comparative analysis of the IO, CoT, and DID prompting methods across three representative models: GPT-3.5 Turbo, GPT-4o, and Claude 3.5 Sonnet. Across all models, the DID framework consistently outperforms both IO and CoT methods, demonstrating its superior capability in handling multi-step reasoning tasks.

For GPT-3.5 Turbo, the DID method achieves an accuracy of 13.27%, significantly outperforming the IO prompt (6.73%) and CoT prompt (8.62%). This result highlights the DID framework's ability to guide the model through structured, step-by-step reasoning, even in less powerful mod-

els like GPT-3.5 Turbo. Traditional prompt methods, such as IO and CoT, tend to struggle with multi-step reasoning because they either rely on direct output generation or follow static, predefined thought chains that lack the flexibility to adapt to more complex scenarios. On GPT-4o, the DID method reaches 70.27%, which far exceeds both IO (43.36%) and CoT (55.85%). The substantial margin between DID and other methods demonstrates its efficacy in managing logical deductions, particularly in more complex reasoning tasks. In contrast, the static nature of IO and the limited flexibility of CoT prevent these methods from fully addressing the intricacies of multi-step deductions. Claude 3.5 Sonnet exhibits the highest overall performance across all methods, with the DID method achieving an impressive accuracy of 89.49%, further extending its lead over IO (74.77%) and CoT (83.68%).

As illustrated in Figure 3, the DID framework excels in progressively guiding LLMs through increasingly complex reasoning steps, especially in tasks where relationships must be deduced from ambiguous information, such as determining family connections. Traditional prompt methods often fail to handle such tasks effectively because they attempt to solve the original complex problem directly, which can result in the model missing critical logical connections. This frequently leads to incorrect conclusions. By breaking down multi-step reasoning problems into simpler subproblems, the DID method ensures that the model remains on track and avoids common pitfalls. This structured approach enables LLMs to generalize effectively and handle more sophisticated logical problems, like those presented in the Alice problem.

## 4.2 MR-GSM8K MATH PROBLEMS

**Task.** MR-GSM8K builds upon the GSM8K benchmark but introduces significantly higher complexity by focusing on meta-reasoning. Since MR-GSM8K not only challenges models to identify and explain errors in provided solutions, but also making it more suitable for assessing the advanced cognitive abilities of LLMs. Only the dataset portion of MR-GSM8K is used, in order to test different methods. And the assessing portion is not used. The dataset includes harder problem types, such as reversed reasoning and programmatic thinking, requiring deeper understanding and reasoning capabilities, thus offering a more rigorous evaluation framework for state-of-the-art models.

**Baseline and Framework Setup.** We compare the performance of the DID framework with the CoT prompt. The CoT is widely used as a method for improving the performance of LLM. All model parameters, including temperature, top-k sampling, and other hyperparameters, are set to their default values for a fair comparison.

**Results** Based on the results shown in 4, the performance comparison across different models (GPT-3.5 Turbo, GPT-4o, and Claude 3.5 Sonnet) for the CoT and DID frameworks reveals the consistent superiority of the DID framework. On GPT-3.5 Turbo, the DID method achieves an accuracy of 73.3%, outperforming CoT (68.1%). Similarly, on GPT-4o, the DID method demonstrates a clear advantage with an accuracy of 83.7%, compared to 82.0% for CoT. For Claude 3.5 Sonnet, the DID method further solidifies its dominance, achieving an accuracy of 92.0%, surpassing CoT (91.3%). This consistent performance across models highlights the effectiveness of the DID approach.

In summary, across GPT-3.5 Turbo, GPT-4o, and Claude 3.5 Sonnet, the DID framework consistently outperforms the CoT framework in terms of accuracy, demonstrating a significant performance advantage. This consistent superiority suggests that the DID method possesses stronger reasoning capabilities and higher reliability for handling complex tasks.

## 4.3 HOLIDAY PUZZLE

**Task.** The *Holiday Puzzle* is a task designed to evaluate the reasoning ability of LLMs in handling holiday arrangements. This task includes 20 examples, each based on a holiday and compensatory workday records from the past decade. The task requires the LLMs to calculate the actual number of extra holiday days people receive, excluding weekends that were already scheduled as rest days.

**Baseline and Framework Setup.** To assess the effectiveness of the DID framework on the holiday calculation problem, we compare it with two alternative prompting methods: the IO prompt (which directly utilizes the LLM without structured prompting) and the CoT prompt. For fairness, all model

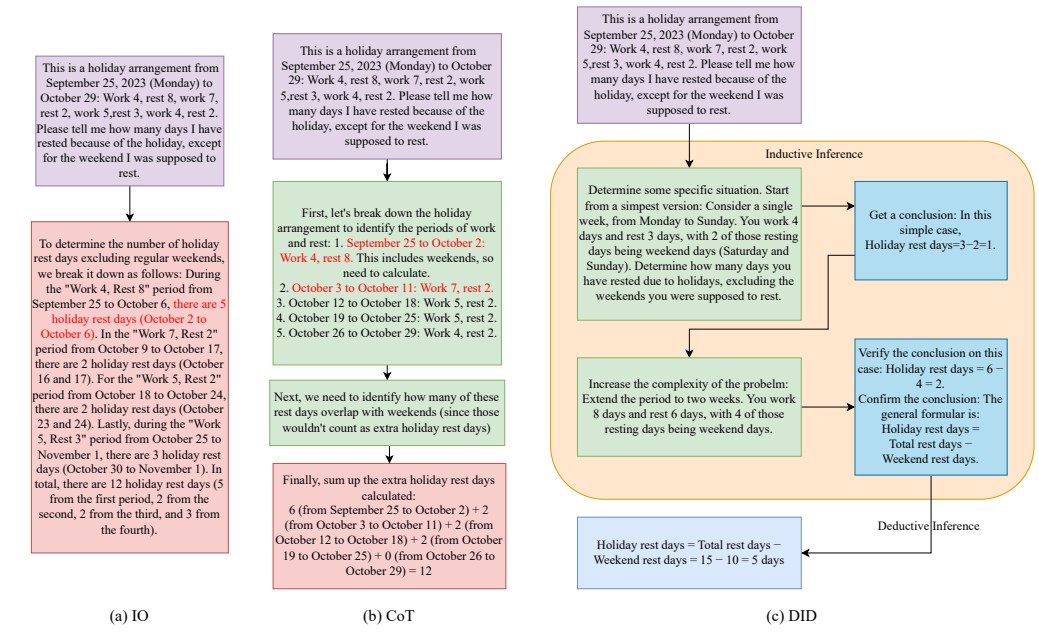

Figure 4: Comparison of reasoning approaches in Large Language Models (LLMs) including the IO method, Chain of Thought (CoT) prompting, and the De-In-Ductive (DID) framework, highlighting the progression from direct output generation to dynamic inductive and deductive reasoning for more adaptive problem-solving.

parameters, including temperature, top-k sampling, and other hyperparameters, are kept at their default values.

**Results**  As shown in 4, the performance comparison across different prompting methods (IO, CoT, and DID) on the "Holiday Puzzle" task highlights the consistent superiority of the DID framework. On GPT-3.5 Turbo, the DID method achieves an accuracy of 5.6%, significantly outperforming IO (0.2%) and CoT (1.4%). This demonstrates that even on less powerful models, the DID framework effectively helps capture the underlying structure of complex temporal and scheduling relationships. Similarly, on GPT-4o, the DID method shows a clear advantage, with an accuracy of 15.4%, compared to 7.8% for IO and 5.2% for CoT. For Claude 3.5 Sonnet, the DID method solidifies its dominance with an accuracy of 24.5%, far surpassing IO (17.4%) and CoT (17.8%).

In this task, the DID framework's stepwise combination of inductive and deductive reasoning proves especially effective for capturing complex patterns such as the relationship *Holiday rest days = Total rest days - Weekend rest days*. Figure 4 demonstrates how DID enables the model to approach this problem systematically, avoiding the risk of falling into incorrect reasoning paths common in traditional prompt methods. By starting with simpler, deductively generated subproblems, the DID framework ensures that the model can handle both the intricacies of date calculations and generalize across varying input conditions. This step-by-step approach allows LLMs to refine their reasoning process as complexity increases, ultimately improving their overall accuracy on tasks like the Holiday Puzzle.

## 5 DISCUSSION

### 5.1 CHALLENGES OF ACHIEVING 100% ACCURACY IN SIMPLE TASKS

Despite the implementation of advanced prompting techniques such as CoT, ToT, and our proposed DID method, LLMs continue to face challenges in consistently achieving 100% accuracy, even on

seemingly simple logical tasks. A key factor underlying this limitation may be the fundamental architecture of LLMs. These models rely on predicting the next token in a sequence, which restricts their ability to maintain a coherent internal representation across multiple reasoning steps. While attention mechanisms allow models to reference previous tokens, they lack the robust cognitive structures that humans use to ensure logical integrity across reasoning processes. Consequently, LLMs can lose track of intermediate steps or overlook crucial logical connections, leading to errors in tasks that might otherwise appear straightforward. This token-based, output-driven mechanism, although effective in many natural language processing tasks, is inherently unsuited for tasks that require rigorous logical consistency and structured reasoning, explaining the persistence of basic mistakes in LLM outputs.

### 5.2 Fine-tuning LLMs with Deductive and Inductive Reasoning

The DID method proposed in this paper primarily focuses on prompting strategies without altering the underlying architecture or fine-tuning the LLM itself. However, future work could investigate the benefits of fine-tuning LLMs on datasets explicitly incorporating deductive and inductive reasoning processes. Fine-tuning strategies such as Reinforcement Learning from Human Feedback (RLHF), Retrieval-Augmented Generation (RAG), or other methods (Lewis et al., 2020; Ouyang et al., 2022; Christiano et al., 2017) could enhance the model's ability to handle complex reasoning tasks. By integrating examples that demonstrate dynamic reasoning adjustments—similar to the DID approach—during the fine-tuning phase, models would gain a deeper understanding of inductive and deductive reasoning patterns. Additionally, emerging techniques like Test-Time Training (TTT) (Sun et al., 2024) could be explored to further improve models' adaptability and reasoning performance during evaluation. These advancements are likely to enhance the consistency and reliability of LLM outputs in structured reasoning tasks by fostering more robust internal representations of logical thought processes.

### 5.3 The ARC Prize Challenge

The ARC (Abstraction and Reasoning Corpus) Prize represents a particularly challenging benchmark designed to evaluate an AI's capacity for abstract reasoning and generalization. Unlike conventional AI tasks that target pattern recognition based on fixed datasets, ARC tests a model's ability to generalize across diverse problem types, making it an ideal platform to assess the adaptability of frameworks such as our De-In-Ductive (DID) method. ARC problems require abstract reasoning across domains like visual pattern recognition and symbolic problem-solving—areas where DID's integration of inductive and deductive reasoning could provide a competitive edge. The DID framework's ability to adapt dynamically based on task complexity, first using inductive reasoning to hypothesize patterns and then applying deductive logic to test them, positions it well for tasks in the ARC benchmark. Given that ARC problems often involve iterative problem-solving and the ability to generalize from minimal examples, we believe the DID method is particularly suited for this task. Future work will focus on evaluating DID's performance on ARC Prize tasks to test its robustness and effectiveness in abstract reasoning scenarios.

## 6 Conclusion

In this work, we introduced the De-In-Ductive (DID) method, a novel framework that dynamically integrates inductive and deductive reasoning to enhance the adaptability and reasoning capabilities of Large Language Models (LLMs). By leveraging cognitive science principles, the DID framework allows LLMs to evolve their problem-solving strategies in response to task complexity, overcoming the rigidity of static prompt structures. Through extensive empirical validation on both standard benchmarks and our custom Holiday Puzzle dataset, we demonstrated significant improvements in accuracy and reasoning quality, achieved without excessive computational costs. However, while DID advances the field, challenges remain in making LLMs more intelligent, particularly in ensuring better generalization to unseen tasks, maintaining adaptability in complex multi-step reasoning, and further refining model biases. Future research must continue to address these issues, paving the way for more robust and cognitively aligned artificial intelligence systems.

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
