# OpenReview forum: "The Role of Deductive and Inductive Reasoning in Large Language Models"
_ICLR.cc/2025/Conference — ICLR 2025 Conference Withdrawn Submission_

### Official Review · Reviewer_2jPs · 2024-10-26

**Soundness:** 2
**Presentation:** 3
**Contribution:** 2
**Rating:** 3
**Confidence:** 3

**Summary:**

The paper introduces the Deductive and InDuctive (DID) method, an approach designed to enhance the reasoning capabilities of Large Language Models (LLMs) by dynamically integrating both deductive and inductive reasoning within the prompt construction process. The DID method is grounded in cognitive science models of human reasoning and aims to mirror human adaptive reasoning mechanisms. The paper empirically validates the efficacy of the DID method on AIW, MR-GSM8K, and Holiday Puzzle, demonstrating significant improvements in solution accuracy and reasoning quality without incurring substantial computational overhead.

**Strengths:**

The paper presents the Deductive and InDuctive (DID) method, which offers an approach to enhancing the reasoning capabilities of Large Language Models (LLMs) by integrating deductive and inductive reasoning within the prompt construction process. This method is inspired by cognitive science, aligning with how humans adapt their reasoning strategies.

**Weaknesses:**

- The insufficiency of the experimental validation is one of the paper's main limitations. The paper only provides empirical validation on three relatively simple datasets and lacks in-depth analysis. On one hand, the authors should consider validating their method on more realistic datasets to demonstrate the effectiveness and generalizability of the proposed approach. On the other hand, benchmarking the DID method against state-of-the-art methods on the same tasks would help position the DID method within the existing landscape of LLM reasoning techniques. This comparison is crucial for establishing the novelty and practical value of the DID framework in enhancing the reasoning capabilities of large language models.

- The paper could benefit from a more detailed comparison with existing methods, such as Tree of Thought (ToT) and Graph of Thought (GoT). It should clearly articulate how it differs and why these differences are significant. Providing a side-by-side comparison with a clear emphasis on the unique aspects of DID would strengthen the novelty argument.

- The paper mentions that the DID approach is inspired by cognitive science models of human reasoning. However, it could provide a deeper dive into the specific cognitive science theories it draws from and how DID reflects these theories. This would help readers understand the theoretical underpinnings better and assess the method's potential for aligning with human-like reasoning.

- The paper claims that the DID method achieves improvements without imposing substantial computational overhead. However, it would be beneficial to see a detailed analysis of the computational costs, including comparisons with other methods on the same level of computational cost (e.g., cot + self-consistent voting), to substantiate this claim.

**Questions:**

Please refer to the Weaknesses.

---

### Official Review · Reviewer_ECwc · 2024-10-29

**Soundness:** 2
**Presentation:** 2
**Contribution:** 3
**Rating:** 3
**Confidence:** 4

**Summary:**

This paper proposes a reasoning framework which dynamically integrates inductive reasoning and deductive reasoning. The proposed framework can adjust its strategy according to the evolving complexity of the problem. The effectiveness has been examined across a diverse set of complex tasks.

**Strengths:**

1. Adaptive reasoning is an important challenge for LLMs. The proposed approach is novel.
2. This paper selects three tasks ranging from various complexity, showing its improvement in flexibility and adaptability of LLMs in complex problem-solving tasks.

**Weaknesses:**

1. Introduction is too long and does not say much.
2. Poor presentation
 - Figure 1 is ambiguous. how complexity is measured? And why models are arranged in x-axis (complexity)?
 - Figure 2 does not help understanding since the concept of "Evolved Hypotheses" in DID part is still too general. Need more information to demonstrate the overall framework. Illustrations of "Knowledge Base", "Extracted Patterns and Facts" seem unnecessary here.
3. Need to improve the orgranization of methodology. Currently it mainly focuses on theoretical analysis, but lack of implementation details. A psuedo code of the algorithm might help readers to understand the overall pipeline.
4. Insufficient experiment baselines. The DID framework is an iterative approach, it's unfair to only compare with prompting method such as IO or CoT. Better to add iterative baselines such as ToT or GoT for a fair comparison.
5. Lack of insights in Discussion section. 5.2 and 5.3 are future works, and 5.1 is kind of speculations, without evidence and in-depth error analysis.

**Questions:**

1. How the hybrid objective function and dynamic adjustment is implemented? From my understanding, there's no "training" process in your setting. how error is calculated and how new instances are selected?
2. What're the prompts for inductive reasoning and deductive reasoning?

---

### Official Review · Reviewer_11fV · 2024-11-01

**Soundness:** 2
**Presentation:** 2
**Contribution:** 1
**Rating:** 3
**Confidence:** 3

**Summary:**

This paper introduces the Deductive and InDuctive(DID) method, which enhances LLM reasoning by dynamically integrating both deductive and inductive reasoning within the prompt construction process. The method is shown to improve over both accuracy and reasoning capabilities on different datasets, without affecting significantly the computational cost.

**Strengths:**

- Integrating both inductive and deductive reasoning within LLM is a very interesting challenge.

**Weaknesses:**

- The introduction is quite long and I think it could be reduced to leave more space to more meaningful details about the method and the experiments.
- The description of the assumptions is neither informally nor formally clear, and some symbols have not been correctly defined (see questions).




OTHER COMMENTS:
- "(..) novel approach designed to enhance LLM reasoning by integrating both inductive and deductive reasoning processes
within the prompt construction framework as the Figure 1 shows." What I understand from Figure 1 is that DID is increasing both the complexity and accuracy wrt IO and CoT methods. However, it is not explained nor what "complexity" refers to in here, nor in which dataset/task the metrics are measured. Hence I found the Figure quite confusing instead of useful.
-"Deductive reasoning, formalized by philosophers like Kant, involves (..)" -> please add a citation here.
-Observed dataset: I think it would me more general to have D_p=\{d_1,...,d_{n_p}\}, similar for the new data, as each problem can have a different number of data.
-"Based on Assumptions 3.1 and 3.1," typo.
- Why the paragraph "Limitations of Chain-of-Thought (CoT) under Assumptions" is in section 3.2? This aspect don't seem to be related to the DID definition. Also, how CoT differs from DID concretly? Is there a fixed alpha, or it relies on a totally different formulation?
- The definition of the complexity is incomplete. What are n_ind and n_ded?
- "The dynamic adjustment enabled by the DID framework, as necessitated by Assumptions 1 and 2, reduces error propagation (..)" Please explain elaborate more this sentence.
- I don't see how the "theoretical adaption analysis" is supported by the description of the framework in Section 3.2.

**Questions:**

1) What is formally the hypothesis H_p? What is the hypothesis space? How the best H is chosen? How does it scale?
2) What is c_0 (eq 3)? It is undefined.
3) What is meant by "new data"? You mean the original available data are split into two different sets, like train and validation, and then used both for the inductive/deductive losses? I don't see otherwise how new (unlabelled/test) data could be used in Eq. 7.
4) How Eq. 8 has been chosen? This means you need to calculate \alpha_t any time a new data is considered?
5) What is the metric reported in Table 1? What is the variance over the 20 runs?
6) What is the role of Eq. 7 as experiments are considered in the zero-shot setting? Also, how the dynamic of data is modeled in the experiments?

---

### Official Review · Reviewer_g9oK · 2024-11-01

**Soundness:** 2
**Presentation:** 1
**Contribution:** 2
**Rating:** 3
**Confidence:** 3

**Summary:**

This paper attempts to introduce a novel approach called DID that uses deduction and induction to solve problems requiring reasoning.

**Strengths:**

1. Flexible approach: While the approach is not entirely novel, it does appear more flexible than existing methods.
2. Improved accuracy: The authors demonstrate emperically that DID method somewhat outperforms the existing CoT methods.

**Weaknesses:**

1. The paper does not contrast this approach with Tree of Thought approaches, or other methods that trade accuracy for inference timesteps. The shown baselines are quite weak and stale at this point in the field.

2. The problems being attacked by this paper would benefit significantly from tool use, which is not a mainstay of the paper. This lack of treatment reduces the impact and novelty of this method.

3. The paper's writing style can be improved to get to the core of the training methodology more quickly. The current structure requires readers to sift through substantial text before reaching key contributions.

**Questions:**

1. Can the authors compare against more relevant baselines including ToT, and possible SoTA methods like o1?
2. AN understanding of how hypothesis can be improved and tailored to the task would be benefical to the application. Can the authors discuss this in more detail as it is central to their paper?

---

### Official Review · Reviewer_j8gM · 2024-11-03

**Soundness:** 2
**Presentation:** 3
**Contribution:** 2
**Rating:** 5
**Confidence:** 3

**Summary:**

This paper presents the De-In-Ductive (DID) prompt strategy, which integrates deductive and inductive reasoning to improve adaptability in large language models (LLMs) for complex reasoning tasks. The DID approach, inspired by cognitive science, dynamically balances inductive hypothesis generation and deductive rule application, allowing models to adjust their reasoning pathways based on task complexity. Empirical validation across benchmark datasets (AIW, MR-GSM8K) and a custom dataset, Holiday Puzzle, highlights reported improvements in solution accuracy and reasoning quality without substantial computational overhead. By combining deductive and inductive reasoning, DID aims to enhance LLM performance in dynamic problem spaces.

**Strengths:**

Originality: The DID framework’s combination of inductive and deductive reasoning is an interesting approach to enhancing reasoning flexibility.

**Weaknesses:**

Clarity of Figures and Descriptions: Some figures (e.g. 1, 4) lack essential information, such as dataset details, metric explanations, and setup descriptions, making it difficult to interpret results.

Prompt Strategy Details: As a prompt strategy paper, the DID method should explicitly define its prompt strategy. Currently, the methodology is insufficiently explained, with few details about how the DID prompts are structured or adapted dynamically.

Reproducibility: The paper lacks reproducibility; results, dataset (Holiday Puzzle), and other experimental resources are not attached.

Baseline Comparisons: Other prompting methods (ToT, T2oT, GoT) are mentioned but not evaluated against, which reduces the paper’s comparative validity.

ARC Challenge Reference: While the DID method is positioned as a potential solution for the ARC Prize Challenge, this claim is not backed by any testing on the ARC tasks, diminishing the value of this comparison.

**Questions:**

Could the authors provide the exact structure of the DID prompts used in experiments?

What steps would be needed to make the experiments reproducible, such as providing the Holiday Puzzle dataset and detailed experimental setup?

Could the authors evaluated against methods like ToT, T2oT, and GoT that were mentioned in the paper?

---

### Note · Authors · 2024-11-26

**Comment:**

Thanks for the detailed reviews!

**Withdrawal Confirmation:**

I have read and agree with the venue's withdrawal policy on behalf of myself and my co-authors.